# Celiac Disease Diagnosed through Screening Programs in At-Risk Adults Is Not Associated with Worse Adherence to the Gluten-Free Diet and Might Protect from Osteopenia/Osteoporosis

**DOI:** 10.3390/nu10121940

**Published:** 2018-12-07

**Authors:** Francesco Tovoli, Giulia Negrini, Vito Sansone, Chiara Faggiano, Teresa Catenaro, Luigi Bolondi, Alessandro Granito

**Affiliations:** 1Unit of Internal Medicine, Department of Medical and Surgical Sciences, University of Bologna, 40136 Bologna, Italy; giu.negrini@gmail.com (G.N.); chiara.fg9@gmail.com (C.F.); teresa.catenaro@studio.unibo.it (T.C.); luigi.bolondi@unibo.it (L.B.); alessandro.granito@unibo.it (A.G.); 2Unit of Gastroenterology, Department of Medical and Surgical Sciences, University of Bologna, 40136 Bologna, Italy; vito.sansone@studio.unibo.it

**Keywords:** celiac disease, gluten, gluten-free diet, screening, outcome, gluten sensitivity, osteoporosis

## Abstract

Screening strategies to detect celiac disease (CD) in at-risk subjects are of paramount importance to prevent the possible long-term complications of this condition. It is therefore of strategic relevance to understand whether patients diagnosed through screening follow a strict gluten-free diet (GFD), as the non-compliance to this diet can make screening efforts pointless. Currently, no studies have verified whether CD patients diagnosed in their adulthood are adhering to the GFD years after the diagnosis. We retrospectively evaluated the medical records of 750 CD patients diagnosed in our center during January 2004–December 2013 to verify differences between screening detected and clinically diagnosed patients. The groups shared a similar adherence to the GFD (91.2 versus 89.8%, *p* = 0.857). Moreover, the rates of non-responsive CD, GFD-induced metabolic alterations, and persistence in controls were also similar. Instead, screening-detected patients had a significantly lower rate of osteopenia/osteoporosis at diagnosis (31.3 versus 46%, *p* < 0.001). In conclusion, screening strategies for CD in at-risk groups should be encouraged even in the adult population. Patients diagnosed through these strategies had no additional problems compared to those diagnosed for clinical suspicion and might benefit from a protective effect against metabolic bone disease.

## 1. Introduction

Celiac disease (CD) is a chronic small intestinal immune-mediated enteropathy precipitated by exposure to dietary gluten in genetically predisposed people [1]. The prevalence of CD is about 1% of the general population [1]. However, a significant underdiagnosis issue exists, and most CD patients are still to be detected [2]. The identification of CD patients relies both on a case-finding strategy in subjects with gluten-related manifestations and on a screening strategy in subjects at high risk of disease. These high-risk populations include first-degree relatives of CD patients, patients with immune deficiencies (Immunoglobulin A deficiency, common variable immunodeficiency), autoimmune diseases (in particular, type 1 diabetes mellitus), congenital chromosomal abnormalities (Down and Turner syndromes) [3,4,5]. Unlike CD patients diagnosed on the basis of the presence of gluten-related symptoms, patients diagnosed through screening often have milder symptoms or no symptoms at all. The gluten-free diet (GFD) has consequently a more limited impact on their symptoms, exposing these patients to a higher theoretical risk of incorrect GFD compliance (and therefore to an increased risk of complications). Therefore, it is of capital importance to verify whether screening-diagnosed patients follow the GFD, as the lack of compliance negates any benefit of an early diagnosis. Unfortunately, until now adherence to the GFD in screening-diagnosed adults has been verified only in two studies performed in Northern Europe cohorts [6,7]. The colleagues found that the adherence of screening-detected subjects was similar to that of patients diagnosed on the basis of clinical suspicion. Other hints came from a recent survey in which 236 patients completed health, quality of life, and dietary adherence questionnaires a median of 18.5 years after childhood diagnosis. Even in this case, the authors reported comparable long-term outcomes between screening-diagnosed and clinically detected patients [8]. However, until now, no study has combined the strengths of a large cohort [7] with an extended follow-up [6], leaving this topic open. Lessons from previous studies in children, in fact, taught us that the compliance may drop after an extended follow-up [9,10], especially in screening-diagnosed subjects [10] and that patients from different geographical regions may have a very different adherence to the GFD [11]. At the same time, not all of the information can be translated from the pediatric experience, as adults may have different peculiarities. For instance, the impact deriving from a significant dietary change after decades of a gluten-containing diet and the increased risk of complicated CD (which is virtually absent in childhood) [12] are to be considered. Moreover, it is in adulthood that CD-related osteoporosis and GFD-induced metabolic alterations may reach their full clinical expression [13].

The objective of our study was to verify whether CD patients diagnosed in adulthood through screening procedures had different characteristics compared to CD patients diagnosed on the basis of the presence of symptoms. In particular, we intended to analyze clinical data at baseline, adherence, and responsiveness to the GFD and metabolic problems induced by the GFD. According to our national guidelines and government policies, every CD patient has to be evaluated six months after the beginning of the GFD and every 18–24 months after that [14]. Consequently, patients followed for a long period can be considered as adhering to the follow-up procedures rather than attention-seeking subjects, thus avoiding (or at least reducing) possible biases. For this reason, we also compared the persistence in the follow-up procedures between groups.

## 2. Materials and Methods

### 2.1. Clinical Setting

We retrospectively analyzed the medical records of patients who were consecutively diagnosed with CD in our outpatient clinic (Bologna Authority Hospital S.Orsola-Malpighi, Bologna, Italy) between January 2004 and December 2013. The final cut-off of December 2013 was chosen to grant a theoretical minimum five-year follow-up even for the most recently diagnosed patients.

### 2.2. Inclusion and Exclusion Criteria

A diagnosis of CD performed according to the North American Society for Pediatric Gastroenterology, Hepatology, and Nutrition [4] was regarded as the key inclusion criterion. Patients with incomplete medical records or unconfirmed diagnosis were excluded from this study. (Figure 1).

All the remaining patients were considered eligible for the following comparative analysis: Clinical presentation (symptomatic versus asymptomatic), severity of histology lesions according to the Marsh–Oberhuber classification (3a versus 3b versus 3c), prevalence of iron-deficiency anaemia, prevalence of osteopenia.

Instead, only patients who performed at least two evaluations (one of which at least six months after the beginning of the GFD) were considered to be eligible for the additional comparative analyses. These further comparisons included: Compliance to the GFD, prevalence of non-responder CD defined according to the Oslo classification [1], and prevalence of metabolic alterations (body weight increase >10 kg, total cholesterol increase >50 mg/dL, development of overt metabolic syndrome).

### 2.3. Clinical Evaluations

Clinical evaluations were scheduled according to the local guidelines (six months after the beginning of the GFD and every 18–24 months after that [14]). Each evaluation consisted of: (1) medical history; (2) physical examination; (3) evaluation of laboratory tests.

Medical history was examined with particular attention to reported intentional or accidental gluten ingestion, modifications of gluten-related symptoms (diarrhea, abdominal pain, bloating, dyspepsia, constipation, skin rash, myalgia, oral aftosis, etc.), onset of any new symptom, and list of current medications [14].

Physical examination included the evaluation of vital signs and body weight. Abdomen, thorax, heart, and neck examinations were also performed [14].

Laboratory tests included: Complete blood count, ferritin, calcium, glycaemia, total cholesterol, alanine and aspartate aminotransferase, thyroid-stimulating hormone, anti-thyroperoxidase and anti-thyroglobulin antibodies, anti-tissue transglutaminase immunoglobulin A, and anti-deamidated gliadin peptides immunoglobulin G antibodies. Additional tests were performed at diagnosis (other anti-organ- and non-organ-specific antibodies) or on a clinical basis (second-level metabolic and osteometabolic tests) [14].

Dual-energy X-ray absorptiometry (DXA) was performed at the diagnosis of CD.

### 2.4. Evaluation of Adherence to the GFD

The patients were considered to have complied to the GFD if all the following criteria were satisfied: (1) No reported intentional or accidental gluten ingestion; (2) absence of CD-related symptoms; (3) negative anti-transglutaminase IgA antibodies [4].

### 2.5. Ethics

This study was approved by the Institutional Review Board of the Bologna Authority Hospital S.Orsola-Malpighi (Protocol 243/2013/O/OssN) and performed according to the Declaration of Helsinki guidelines. All patients signed an informed consent.

### 2.6. Sample Size Considerations

The main aim of our study was to compare the rate of patients strictly following the GFD during the follow-up. On the basis of the previous reviews on the adult population [6,7], we estimated a global adherence to the GFD of 90%. To detect a decrease of 10% in one of the two study groups with an alpha error of 0.05 and a power of 90%, we estimated to enroll a minimum of 640 patients.

### 2.7. Statistical Analysis

Distribution of continuous variables was assessed with a Shapiro–Wilk test, which showed non-normal distributions. Consequently, continuous variables were expressed as median and interquartile range. Categorical variables were expressed as frequencies. Group comparisons were subsequently performed using the Mann–Whitney test for continuous variables and the two-tailed Fisher’s test for categorical variables. Binary logistic regression was performed to assess the relationship between correct compliance with the GFD (dependent variable) and other clinical variables. Log-rank test and Cox proportional hazard models were used to evaluate the relationship between the persistence to the follow-up procedure and other clinical variables of interest. Variables for which the association in the univariate analysis was *p* < 0.10 were entered into the multivariate models. A *p* < 0.05 was considered to be the cut-off for statistical significance. All of the statistical analyses were performed with SPSS version 23.0 (SPSS Inc., Chicago, IL, USA).

## 3. Results

### 3.1. Study Population

A total of 750 CD patients were identified using the specified inclusion and exclusion criteria. Of these patients, 115 (15.3%) had been identified through screening procedures, and 635 (84.7%) because of clinical suspicion. The clinical motivations leading to the screening procedures are reported in Table 1.

### 3.2. Characteristics at Diagnosis

A comparison of the main characteristics of screening- and clinically detected patients is detailed in Table 2.

Female sex was slightly preponderant in clinically detected patients, while the age at the diagnosis did not significantly differ between groups. As expected, the prevalence of symptomatic patients was sharply higher in clinically detected compared to screening-detected patients. In parallel, iron-deficiency anemia and osteopenia/osteoporosis were also more prevalent in the clinically detected group of patients. No differences in the severity of villous atrophy were noted.

### 3.3. Follow-Up

A total of 13 patients in the screening group (11.3%) and 74 clinically detected patients (11.7%) dropped out after the first evaluation in our centre (*p* = 1.000). The remaining patients (102 and 561, respectively) were assessed for compliance to the GFD, prevalence of non-responder CD, and metabolic alterations (Table 3).

Compliance to the GFD was similar in screening- and clinically detected patients (91.2 versus 89.8%, *p* = 0.857). Two patients in the screening-diagnosed group and 25 patients in the clinically detected group referred possible accidental contaminations with gluten, mainly when they ate at work or at a restaurant. Instead, seven and 32 patients, in the screening- and clinically-detected groups respectively, voluntarily ingested gluten at least twice per month. At the binary logistic regression analysis, only the presence of gluten-related symptoms at baseline was marginally associated with better compliance (hazard ratio 2.110, 95% confidence interval 0.888–5.012, *p* = 0.091). On the contrary, sex (*p* = 0.283), age at the diagnosis (*p* = 0.504), and presence of a relative with CD (*p* = 0.183) did not correlate with compliance.

Overall, persistent symptoms were found in 19 (18.6%) screening-diagnosed patients and 153 (27.3%) clinically detected patients. Briefly, the aforementioned incomplete compliance to the GFD was one of the leading causes of persistent symptoms in both groups. Co-existing irritable bowel syndrome and gastroesophageal reflux disease were not significantly different across the study groups. Complicated CD was confirmed to be a rare condition. Interestingly, all seven cases of complicated CD were found in the clinically diagnosed patients and not in the screening-diagnosed group. Complications were found at diagnosis in three cases (two small bowel adenocarcinoma, one enteropathy-associated T cell lymphoma) and were diagnosed in the follow-up in the remaining five cases (three patients with type-1 refractory CD, one case of a type-2 refractory CD, and one case of hyposplenism).

Persistence in the follow-up procedures was also similar between the two groups, with a median follow up of 6.5 years (95% confidence interval 5.1–7.9) in screening-detected and 6.3 years (95% confidence interval 5.7–6.9) in clinically detected patients (*p* = 0.452) (Figure 2).

Age at diagnosis was the only factor associated with persistence, with an inverse correlation (hazard ratio 0.988, 95% confidence interval 0.982–0.994, *p* < 0.001). Sex (*p* = 0.325), familiarity for CD (*p* = 0.696), symptoms at the diagnosis (*p* = 0.155), and adherence to GFD (*p* = 0.392) were not associated with the length of the follow-up.

## 4. Discussion

In this paper, we assessed different clinical aspects of adult CD patients in which their condition was discovered thanks to screening programs. In particular, we explored both their baseline and their follow-up evaluations. These tasks were performed in a large cohort with a median six-year follow-up.

First, our data showed that adult CD patients diagnosed by screening had a GFD adherence which was similar to that of clinically detected patients after a six-year median follow-up. Both groups had an adherence as high as 90%. As previously stated, local policies may influence compliance. In the case of this study, it should be noted that Italy has official government policies protecting CD patients and helping them in the management of their conditions. These policies establish that diagnosed CD patients receive vouchers to buy specially produced gluten-free foods for up to 120 euro/month and call for strict respect of the local guidelines which recommend a clinical and laboratory follow-up at regular intervals. It is, therefore, possible that even patients who are diagnosed in the absence of severe symptoms are sufficiently eased in their difficult endeavors, which would favor higher compliance. The similarity in the compliance to the GFD of screening- and clinically detected CD patients had been already suggested in a series of studies on pediatric populations, mainly from North-European countries [15,16,17]. Data in the adult screening-detected population are more limited. Viljamaa and colleagues [6] firstly reported adherence of 82 versus 77% in 53 screening-detected and 44 clinically detected adult patients after a 14-year follow-up. In the only large study, Ukkola et al. [7] analyzed 123 screening-detected patients versus 698 patients diagnosed because of clinical suspicion. The self-reported adherence to the GFD after one year was similar between the study groups (91 versus 85%). Compared to Ukkola et al., the design of our study is different. While we lack the strengths of a prospective evaluation, we benefited from a longer follow-up and a physician-assessed adherence. Moreover, a difference in the geographical regions of enrolment should also be considered. With all these differences in mind, our results are surprisingly similar to those of Ukkola, thus validating them even in a different social context and in a more extended follow-up period. We also report novel findings based on the analysis of the persistence in the follow-up procedures, the rate of the non-responsive CD, and the metabolic alterations potentially induced by the GFD. In particular, all these factors were consistent across the study groups. It is also interesting to note that all the patients with complications belonged to the clinically detected group, even if our study was not designed nor powered enough to detect significant differences in this variable. A final word on this topic, therefore, will only come from extensive studies with decade-long follow-up.

Second, our data validated on a large scale the preliminary evidence that screening-detected patients have peculiar features during diagnosis when compared to patients recognized because of clinical suspicion. Approximately 25% of our screening-detected patients had mild gluten-related symptoms which could have led to the diagnosis, suggesting that the local screening strategies were reasonably good. This rate is, in fact, lower if compared to those previously described in adults [18] and pediatric series [15], in which at least half of the cases diagnosed by screening had symptoms. More interestingly, our screening-diagnosed patients were significantly less affected by iron-deficiency anemia and osteopenia. In a series of 19 screening-detected patients, Mustalahti and colleagues [19] firstly reported that bone mineral density (BMD) was lower than average. Following this report, Sundar et al. [20] compared the characteristics of 24 screening-detected versus 105 clinically detected CD patients, confirming a higher rate of BMD abnormalities in the latter group but also calling for further evidence from larger cohorts. We, therefore, suggest that the prevention of metabolic bone disease should strongly encourage CD screening in at-risk subjects. In fact, CD subjects are at increased risk of fractures [21,22] and benefit from GFD, which generally leads to an improvement in BMD in the first 12 months [23,24].

The strengths of our study include a large cohort of patients, the systematic clinical assessment of the compliance to the GFD, and the availability of additional information including the rate of clinical response to the GFD and its metabolic impact. We are aware that our study also comes with limitations, including its retrospective nature, the lack of systematic assessment of all features of metabolic syndrome (for instance, high-density lipoprotein cholesterol), and the lack of structured questionnaires for an even more comprehensive collection of both symptoms and dietary adherence. Finally, because of the observational nature of our study, we did not investigate the prevalence of some polymorphisms involved in iron absorption in children (DMT 1 IVS4 + 44C > A) [25] and lower bone mineral density in adults (IL1B-511T) [26], as reported in previous studies in CD patients.

## 5. Conclusions

Our data support the use of screening procedures for CD in the adult population. In particular, we demonstrated that an early diagnosis might protect from severe metabolic bone disease. Also, compared to clinically diagnosed patients, screening-diagnosed patients did not show an impaired adherence to the GFD nor an increased rate of GFD-related problems.

## Figures and Tables

**Figure 1 nutrients-10-01940-f001:**
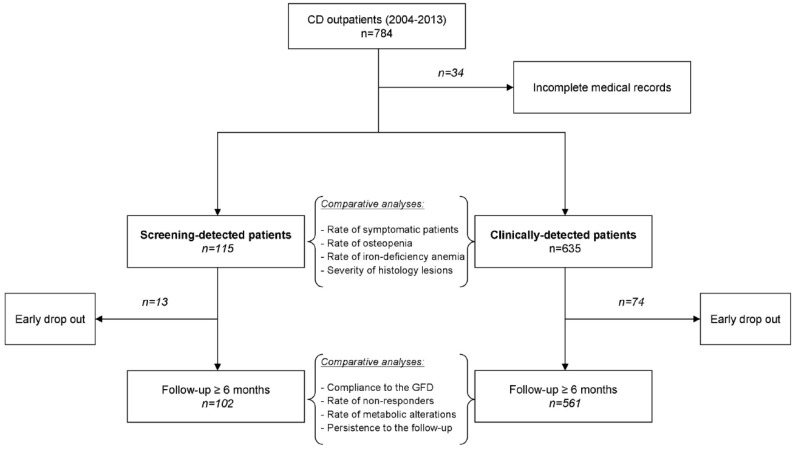
Patients’ flow chart. CD: celiac disease

**Figure 2 nutrients-10-01940-f002:**
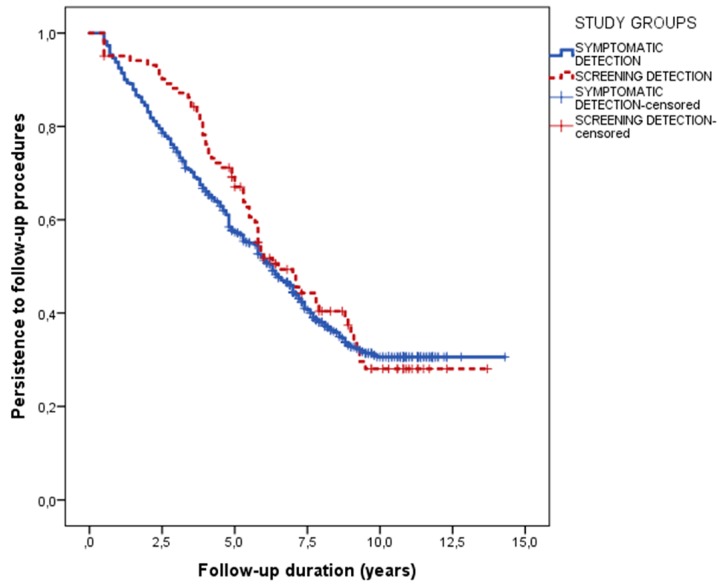
Persistence in the follow-up procedures in patients detected by screening and in patients diagnosed because of clinical suspicion.

**Table 1 nutrients-10-01940-t001:** Conditions which justified the screening for celiac disease in at-risk groups.

**Familiar Screening (Total)**	**78 (67.8)**
Index case: brother/sister	26 (22.6)
Index case: father/mother	18 (15.7)
Index case: son/daughter	18 (15.7)
Index case: nephew/grandchild	11 (9.6)
Index case: cousin	5 (4.3)
**At-Risk Associated Conditions (Total)**	**37 (32.2)**
Autoimmune thyroid disease	10 (8.7)
Type-1 diabetes mellitus	6 (5.2)
Primary biliary cholangitis	3 (2.6)
IgA deficiency	3 (2.6)
Down syndrome	3 (2.6)
Vitiligo	3 (2.6)
Sjogren syndrome	1 (0.9)
Turner syndrome	1 (0.9)
Other	7 (6.1)

Data are reported as absolute frequencies (percentage).

**Table 2 nutrients-10-01940-t002:** Characteristics of screening- and clinically detected patients.

Parameter	Screening (*n* = 115)	Clinical Suspicion (*n* = 635)	*p*
Age (Years)	34 (22–46)	33 (22–44)	0.690
Sex (Female)	80 (69.6)	502 (79.1)	0.021
Time since Diagnosis (Years)	1.5 (0.0–5.0)	2.0 (0.0–6.50)	0.145
Symptoms	29 (25.2)	594 (93.5)	<0.001
Iron-Deficiency Anemia	25 (21.7)	324 (51.0)	<0.001
Osteopenia/Osteoporosis	36 (31.3)	292 (46.0)	0.003
Histology			
-Marsh 3a	33 (28.7)	182 (28.7)	
-Marsh 3b	43 (37.4)	180 (28.3)	0.100
-Marsh 3c	39 (33.9)	273 (42.3)	

Categorical variables are reported as absolute number (percentage), continuous variables are described as median (interquartile range).

**Table 3 nutrients-10-01940-t003:** Comparison of the compliance to the gluten-free diet, persisting symptoms, and metabolic alterations at the follow-up evaluations between screening- and clinically detected patients.

Parameter	Screening (*n* = 102)	Clinical suspicion (*n* = 561)	*p*
Correct Compliance	93 (91.2)	504 (89.8)	0.857
Accidental Contaminations	2 (2.0)	25 (4.5)	0.410
Voluntary Gluten Ingestion	7 (6.9)	32 (5.7)	0.647
Gerd-like Symptoms	2 (2.0)	17 (3.0)	0.753
Ibs-like Symptoms (Total)	8 (7.8)	79 (14.1)	0.110
(a) classical ibs	4 (3.9)	43 (7.7)	0.212
(b) Diarrhea-predominant IBS	1 (1.0)	6 (1.1)	1.000
(c) Constipation-predominant IBS	3 (2.9)	30 (5.3)	0.457
Metabolic Alterations (tOtal)	15 (14.7)	96 (17.1)	0.666
(a) weight increase >10%	7 (6.9)	34 (6.1)	0.823
(b) cholesterol increase >50 mg/dL	7 (6.9)	36(6.4)	0.864
(c) metabolic syndrome	1 (1.0)	26 (4.6)	0.103

IBS: irritable bowel syndrome; GERD: gastroesophageal reflux disease. Categorical variables are reported as absolute number (percentage), continuous variables are described as median (interquartile range).

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
