# Peer review of "Celiac Disease Diagnosed through Screening Programs in At-Risk Adults Is Not Associated with Worse Adherence to the Gluten-Free Diet and Might Protect from Osteopenia/Osteoporosis"

_nutrients, 2018, doi:10.3390/nu10121940_

Round 1
Reviewer 1 Report
The section "materials and methods" must be clarified by including a more detailed description of how the researcher measured different variables (e.g. adherence, symptoms...).
Line 86. We recommend to improve the descriptions of the evaluations. What did these evaluations consist of?
Table 1 can be redesigned to look clearer.
Check the use of capital letters in Figure 1 and FIGURE 2
It would be interesting to include in table 2 the variable "time since diagnosis".
Table titles can be shorter and clearer and use the table foot for clarifications.
Table 3 must be redesign to help interpretation.
Line 224: Conclusions can be rewritten to be clearer.
This study is of scientific interest but some aspects can be clarify.
Author Response
We would like to thank the reviewer for his/her suggestions. We addressed all of the raised points and we feel that the reviewer's suggestions have helped us in improving the quality of our manuscript. However, we are at disposal of the reviewer, should he/she think that any other modification is needed to further improve our paper.
Our point-to-point response can be found in the attached file. Modifications have been tracked in the paper using a dark blue font.
______________________________________________________________________________
Response to Reviewer 1 Comments
Point 1: The section "materials and methods" must be clarified by including a more detailed description of how the researcher measured different variables (e.g. adherence, symptoms...).
Response 1: We thank the reviewer for this suggestion. We re-organized the “materials and methods” section, creating two new paragraphs titled “2.3 - clinical evaluations” and “2.4 – evaluation of adherence to the GFD”.
Point 2: Line 86. We recommend to improve the descriptions of the evaluations. What did these evaluations consist of?
Response 2: From the previous response, we now report more in detail the evaluations (which were performed in accordance to our local guidelines:
2.3 Clinical evaluations
Clinical evaluations were scheduled according to the local guidelines (six months after the beginning of the GFD and every 18-24 months after that [13]). Each evaluation consisted of: (1) medical history; (2) physical examination; (3) evaluation of laboratory tests.
Medical history was taken with particular attention to: reported intentional or accidental gluten ingestion, modifications of gluten-related symptoms (diarrhoea, abdominal pain, bloating, dyspepsia, constipation, skin rash, myalgia, oral aftosis, etc.), onset of any new symptom, and list of current medications [13].
Physical examination included evaluation of vital signs and body weight. Abdomen, thorax, heart, and neck examinations were also performed [13].
Laboratory tests included: complete blood count, ferritin, calcium, glycaemia, total cholesterol, alanine and aspartate aminotransferase, thyroid stimulating hormone, anti-thyroperoxidase and anti-thyroglobulin antibodies, anti-tissue transglutaminase IgA and anti-deamidated gliadin peptides IgG antibodies. Additional tests were performed at the diagnosis (other anti-organ and non-organ specific antibodies) or on a clinical basis (second level metabolic and ostheometabolic tests) [13].
Dual-energy X-ray absorptiometry (DXA) was performed at the diagnosis of CD.
We also added a paragraph dedicated to the definition of adherence to the GFD. In detail, adherence was evaluated on a combination of medical history and serology, as recommended by the NASPGHAN guidelines
2.4 Evaluation of adherence to the GFD
The patients were considered to have complied to the GFD if all the following criteria were satisfied: (1) no reported intentional or accidental gluten ingestion; (2) absence of CD‐related symptoms; (3) negative anti‐transglutaminase IgA antibodies [4].
Point 3: Table 1 can be redesigned to look clearer.
Response 3: We agree that Table 1 might have been confusing, especially in the part of the family screening. The Table was redesigned to make more clear these points: (1) the bold titles in capital letters (family screening, associated conditions) report the total number of cases for these two most frequent sitatuations; (2) the part of the familiar screening now clearly reports which familiars were the index-cases prompting a screening procedure in the patient.
FAMILIAR SCREENING (TOTAL) | 78 (67.8) |
Index case: brother/sister | 26 (22.6) |
Index case: father/mother | 18 (15.7) |
Index case: son/daughter | 18 (15.7) |
Index case: nephew/grandchild | 11 (9.6) |
Index case: cousin | 5 (4.3) |
AT-RISK ASSOCIATED CONDITIONS (TOTAL) | 37 (32.2) |
Autoimmune thyroid disease | 10 (8.7) |
Type-1 diabetes mellitus | 6 (5.2) |
Primary biliary cholangitis | 3 (2.6) |
IgA deficiency | 3 (2.6) |
Down syndrome | 3 (2.6) |
vitiligo | 3 (2.6) |
Sjogren syndrome | 1 (0.9) |
Turner syndrome | 1 (0.9) |
Other | 7 (6.1) |
Point 4: Check the use of capital letters in Figure 1 and FIGURE 2
Response 4: Thanks for pointing out this discrepancy, now both TABLE and FIGURE have been edited using capital letters
Point 5: It would be interesting to include in table 2 the variable "time since diagnosis".
Response 5: We agree with this suggestion. Table 2 now includes the suggested variable, which was not significantly different between the study groups.
SCREENING (N=115) | CLINICAL SUSPICION (N=635)
| P | |
AGE (YEARS) | 34 (22-46) | 33 (22-44) | 0.690 |
SEX (FEMALE) | 80 (69.6) | 502 (79.1) | 0.021 |
TIME SINCE DIAGNOSIS (YEARS) | 1.5 (0.0-5.0) | 2.0 (0.0-6.5) | 0.145 |
SYMPTOMS | 29 (25.2) | 594 (93.5) | <0.001< span=""> |
IRON DEFICIENCY ANEMIA | 25 (21.7) | 324 (51.0) | <0.001< span=""> |
OSTEOPENIA/OSTEOPOROSIS | 36 (31.3) | 292 (46.0) | 0.003 |
HISTOLOGY | |||
- Marsh 3a | 33 (28.7) | 182 (28.7) | |
- Marsh 3b | 43 (37.4) | 180 (28.3) | 0.100 |
- Marsh 3c | 39 (33.9) | 273 (42.3) |
Point 6: Table titles can be shorter and clearer and use the table foot for clarifications.
Response 6: We thanks the reviewer for this suggestion. Now we report the clarifications in the table foot. Also, most titles have been shortened and made more clear.
Point 7: Table 3 must be redesign to help interpretation.
Response 7: We apologize for the difficult interpretation. We now tried to make the Table more easy to digest using capital and non-capital letters. In particular, subtypes of IBS and metabolic alterations are now described using non-capital letters.
SCREENING (N=102) | CLINICAL SUSPICION (N=561) | P | |
CORRECT COMPLIANCE | 93 (91.2) | 504 (89.8) | 0.857 |
ACCIDENTAL CONTAMINATIONS | 2 (2.0) | 25 (4.5) | 0.410 |
VOLUNTARY GLUTEN INGESTION | 7 (6.9) | 32 (5.7) | 0.647 |
GERD-LIKE SYMPTOMS | 2 (2.0) | 17 (3.0) | 0.753 |
IBS-LIKE SYMPTOMS (TOTAL) | 8 (7.8) | 79 (14.1) | 0.110 |
a) Classical IBS | 4 (3.9) | 43 (7.7) | 0.212 |
b) Diarrhoea predominant IBS | 1 (1.0) | 6 (1.1) | 1.000 |
c) Constipation predominant IBS | 3 (2.9) | 30 (5.3) | 0.457 |
METABOLIC ALTERATIONS (TOTAL) | 15 (14.7) | 96 (17.1) | 0.666 |
a) weight increase >10% | 7 (6.9) | 34 (6.1) | 0.823 |
b) cholesterol increase >50 mg/dl | 7 (6.9) | 36(6.4) | 0.864 |
c) metabolic syndrome | 1 (1.0) | 26 (4.6) | 0.103 |
Point 8: Line 224: Conclusions can be rewritten to be clearer.
Response 8: Thanks for pointing out this issue. The final sentences have been shortened to deliver a more concise and clear message to the readers.
5. Conclusions
Our data support the use of screening procedures for CD in the adult population. In particular, we demonstrated that an early diagnosis might protect from severe metabolic bone disease. Also, compared to clinically-diagnosed patients, screening-diagnosed patients did not show an impaired adherence to the GFD nor an increased rate of GFD-related problems.

Reviewer 2 Report
The paper is novel and findings are remarkable. However, major revision applies due to the following points:
Introduction
-Inappropriate and old-fashioned definition for CD, not even aligned to the reference quoted
- Paper PMID 30228890 has great implications in the background of this study and it should be cited.
Methods:
The author stated: "Continuous variables are expressed as a mean and SD, or as a median and interquartile range, as appropriate".
I do not see means and standard deviation in the result section. Which data showed normality and were considered as means + SD? If so, which test has been applied to assess normality of the distribution? Please state these points in the method section.
Again: "Group comparisons were subsequently performed with unpaired Student's t‐tests for normally distributed variables, and the Mann–Whitney or Kruskall–Wallis test for non‐normally distributed variables".
As above, which test has been applied to assess normality of the distribution? Which data were normal and thus expressed as means?
Please state these points in the method section.
-Discussion:
Line 202 Please correct "decades-long follow-up" with "decade-long follow-up"
Author Response
We would like to thank the reviewer for his/her suggestions. We addressed all of the raised points and we feel that the reviewer's suggestions have helped us in improving the quality of our manuscript. However, we are at disposal of the reviewer, should he/she think that any other modification is needed to further improve our paper.
Our point-to-point response can be found in the attached file. Modifications have been tracked in the paper using a red font. ____________________________________________________________________________
Response to Reviewer 2 Comments
Point 1: Inappropriate and old-fashioned definition for CD, not even aligned to the reference quoted
Response 1: We agree with the reviewer. We aligned the text according to the Oslo definitions for celiac disease and related terms:
“Celiac disease (CD) is a chronic small intestinal immune-mediated enteropathy precipitated by exposure to dietary gluten in genetically predisposed people [1].”
Point 2: Paper PMID 30228890 has great implications in the background of this study and it should be cited.
Response 2: We thank the reviewer for providing us this fresh reference. We are sorry if we overlooked it in our literature research, which began earlier this year. We agree that the suggested paper has implications in the background and therefore we added it to the reference with a dedicated sentence in the Introduction
“Other hints came from a recent survey, in which 236 patients completed health, quality of life and dietary adherence questionnaires a median of 18.5 years after childhood diagnosis. Even in this case, the Authors reported comparable long-term outcomes between screening- and clinically-detected patients [8].”
Point 3: The author stated: "Continuous variables are expressed as a mean and SD, or as a median and interquartile range, as appropriate".I do not see means and standard deviation in the result section. Which data showed normality and were considered as means + SD? If so, which test has been applied to assess normality of the distribution? Please state these points in the method section.
Response 3: Thanks for pointing out this issue. We referenced to the possibility of having both normally and non-normally distributed variables in our original draft. Actually, when the Shapiro-Wilk test was performed, all of the main variables were non-normally distributed. For this reason, the reviewer did not find any means + SD. We agree that the current formulation is redundant and potentially misleading. We modified accordingly this part of the Methods:
“Distribution of continuous variables was assessed with a Shapiro-Wilk test, which showed non-normal distributions. Consequently, continuous variable are expressed as median and interquartile range. Categorical variables are expressed as frequencies. Group comparisons were subsequently performed using Mann–Whitney test for continuous variables and the two‐tailed Fisher's test for categorical variables.”
Point 4: Again: "Group comparisons were subsequently performed with unpaired Student's t‐tests for normally distributed variables, and the Mann–Whitney or Kruskall–Wallis test for non‐normally distributed variables".As above, which test has been applied to assess normality of the distribution? Which data were normal and thus expressed as means? Please state these points in the method section.
Response 4: Please refer to the response to the previous point.
Point 5:-Discussion: Line 202 Please correct "decades-long follow-up" with "decade-long follow-up"
Response 5: Thanks for pointing out this typo. We now fixed the error.

Reviewer 3 Report
This is an excellent paper and fills a gap in the literature which tends to focus on the biopsy as the gold standard. You have demonstrated that the purpose of screening is not merely to rule out disease and gives us insight into the variation and health transitions within this population. These findings have both clinical and health policy implications.
Your article brings a fresh perspective to the use of screening to identify people with celiac disease who where symptoms are silent. It leads me to ask whether some of your "silent" patients possess one or more polymorphisms involved in iron and/or calcium absorption?
As you may know there is some evidence that for iron absorption in children (DMT 1 IVS4+44C>A) , and lower bone mineral density found in adults carrying the T allele of the interleukin-1β gene (IL1B-511T). Since your clinic follows patients overtime, is there a possibility of a genetic study in this population?
Tolone C, Bellini G, Punzo F, et al. The DMT1 IVS4+44C>A polymorphism and the risk of iron deficiency anemia in children with celiac disease. PloS one. 2017;12(10):e0185822. PMID:10.1371/journal.pone.0185822.
Moreno ML, Crusius JB, Chernavsky A, et al. The IL-1 gene family and bone involvement in celiac disease. Immunogenetics. 2005;57(8):618-620. PMID:10.1007/s00251-005-0033-x.
Author Response
Thanks for your kind comments. We provided our replies in the attached file.
Modifications have been reported in the text using a dark green font.
____________________________________________________________________________
Response to Reviewer 3 Comments
Point 1: Your article brings a fresh perspective to the use of screening to identify people with celiac disease who where symptoms are silent. It leads me to ask whether some of your "silent" patients possess one or more polymorphisms involved in iron and/or calcium absorption?
As you may know there is some evidence that for iron absorption in children (DMT 1 IVS4+44C>A) , and lower bone mineral density found in adults carrying the T allele of the interleukin-1β gene (IL1B-511T). Since your clinic follows patients overtime, is there a possibility of a genetic study in this population?
Tolone C, Bellini G, Punzo F, et al. The DMT1 IVS4+44C>A polymorphism and the risk of iron deficiency anemia in children with celiac disease. PloS one. 2017;12(10):e0185822. PMID:10.1371/journal.pone.0185822.
Moreno ML, Crusius JB, Chernavsky A, et al. The IL-1 gene family and bone involvement in celiac disease. Immunogenetics. 2005;57(8):618-620. PMID:10.1007/s00251-005-0033-x.
Response 1: We are grateful to the reviewer for his/her considerations. Our study was observational in its design, so we did not collect samples for genetic studies. However, we acknowledge that the reviewer’s suggestion is very valid indeed and could be the base for future studies in our populations.We also added the provided references, as well as a special statement when reporting limitations of our study:
“Finally, due to the observational nature of our study, we did not investigate the prevalence of some polymorphisms involved in iron absorption in children (DMT 1 IVS4+44C>A) [25] and lower bone mineral density in adults (IL1B-511T) [26], as reported in previous studies in CD patients."

Round 2
Reviewer 2 Report
Paper improved. All issues addressed